# Concealed Substrates in Brugada Syndrome: Isolated Channelopathy or Associated Cardiomyopathy?

**DOI:** 10.3390/genes13101755

**Published:** 2022-09-28

**Authors:** Chiara Di Resta, Jan Berg, Andrea Villatore, Marianna Maia, Gianluca Pili, Francesco Fioravanti, Rossella Tomaiuolo, Simone Sala, Sara Benedetti, Giovanni Peretto

**Affiliations:** 1Genomic Unit for the Diagnosis of Human Pathologies, IRCCS San Raffaele Scientific Institute, 20132 Milan, Italy; 2School of Medicine, Vita-Salute San Raffaele University, 20132 Milan, Italy; 3Department of Cardiac Electrophysiology and Arrhythmology, IRCCS San Raffaele Scientific Institute, 20132 Milan, Italy; 4UOC Screening Neonatale e Malattie Metaboliche, ASST Fatebenefratelli Sacco Ospedale dei Bambini “Vittore Buzzi”, 20157 Milan, Italy

**Keywords:** Brugada syndrome, substrate, cardiomyopathy, sudden cardiac death, inflammation, fibrosis, genetics, ventricular arrhythmia

## Abstract

**Simple summary:**

Having a new and more precise definition of BrS, based on its cardiomyopathic component, may be crucial to meliorate the current clinical management of patients, at (i) diagnostic, (ii) prognostic, and (iii) therapeutic levels: (i) diagnostic, since specific tests may be added to the current standards of BrS to identify associated arrhythmogenic substrates; (ii) prognostic, since multiple factors from an extended diagnostic workup may be associated with an increased arrhythmic risk (as already demonstrated in many cardiomyopathies), subsequently improving the patient selection for a primary prevention ICD implant; (iii) at therapeutic levels, since the identification of unexpected substrates may turn into a significant change in the current treatment practice.

**Abstract:**

Brugada syndrome (BrS) is an inherited autosomal dominant genetic disorder responsible for sudden cardiac death from malignant ventricular arrhythmia. The term “channelopathy” is nowadays used to classify BrS as a purely electrical disease, mainly occurring secondarily to loss-of-function mutations in the α subunit of the cardiac sodium channel protein Nav1.5. In this setting, arrhythmic manifestations of the disease have been reported in the absence of any apparent structural heart disease or cardiomyopathy. Over the last few years, however, a consistent amount of evidence has grown in support of myocardial structural and functional abnormalities in patients with BrS. In detail, abnormal ventricular dimensions, either systolic or diastolic dysfunctions, regional wall motion abnormalities, myocardial fibrosis, and active inflammatory foci have been frequently described, pointing to alternative mechanisms of arrhythmogenesis which challenge the definition of channelopathy. The present review aims to depict the status of the art of concealed arrhythmogenic substrates in BrS, often resulting from an advanced and multimodal diagnostic workup, to foster future preclinical and clinical research in support of the cardiomyopathic nature of the disease.

## 1. Brugada Syndrome: Definition and Current Classification

Brugada syndrome (BrS) is an inherited autosomal dominant genetic disorder, first described in 1992 [1], which combines typical electrocardiographic findings with an increased risk of malignant ventricular arrhythmias. Its prevalence is estimated from 1 in 5000 to 1 in 2000 cases, with a strong male predominance [2].

Current international guidelines [3,4] agree in defining BrS in presence of a type 1 Brugada electrocardiogram (ECG) pattern, i.e., a persistent ST-segment elevation ≥2 mm followed by a negative T-wave in ≥1 of the right precordial leads V1 to V2, occurring either spontaneously or following a sodium channel blocker test (Figure 1). However, according to the Shanghai score system of 2016 [5], if the type 1 pattern is unmasked during a sodium channel blocker test, then clinical history, family history, and a genetic test need to be evaluated to meet the diagnostic criteria.

Despite recent advances, the pathogenetic mechanisms of the disease remain not fully understood. BrS had been initially proposed to be a primary disease functionally involving impairments in the electric potential transmission. BrS was defined as a channelopathy, due to the association of the disease genotype with loss-of-function mutations in genes encoding subunits of the cardiac ion channels [6]. A consistent amount of attention was invested in mutations in the *SCN5A* gene, encoding the α subunit of the cardiac sodium channel protein Nav1.5, responsible for the initial upstroke of the action potential [7]. This had been thought to happen in the absence of ischemia, electrolyte disturbance, or structural heart disease, as supported by silent imaging and post-mortem pathology [8].

Nevertheless, several studies suggested that subtle structural or microscopic abnormalities may actually take place in BrS, including dilation of the right ventricular outflow tract (RVOT), localized inflammation, and fibrosis [9,10]. These observations lead to a rethink of the context of the disease, referring it to apparently normal hearts instead of structurally normal hearts, paving the way for a controversial overlap between BrS and cardiomyopathies [11,12]. Indeed, case reports and case series exploring the presence of concealed substrates in BrS are still preliminary.

The disclosure of concealed substrate abnormalities in BrS may be the answer to the perception of BrS as more than a pure channelopathy, potentially enabling an improvement in the current diagnostic, prognostic, and therapeutic workflow. The present review aims at exploring this concept while providing an updated description of cardiomyopathic changes associated with the disease, from pathophysiological, diagnostic, and prognostic viewpoints.

## 2. Inheritance and Genetic Bases of Brugada Syndrome: The State of The Art

At the time that the first genetic alteration in the *SCN5A* gene underlying BrS was reported in 1998 [6], highlighting an autosomal dominant inheritance, two other BrS genetic hallmarks had already been recognized: incomplete penetrance and variable expressivity. To date, more than 150 loss-of-function alterations have been described in the *SCN5A* gene [13], leading to a decrease in the I-Na^+^ current and a consequent shortening of the depolarization phase of the action potential [7].

Some studies suggested a role for *SCN5A* alterations in the prediction of patients’ arrhythmic risk. Indeed, carriers of a deleterious variant in the *SCN5A* gene show a spontaneous BrS ECG [14] and a more aggressive arrhythmic phenotype; however, this feature needs to be further investigated [15].

Overall, about 20% of cases are caused by rare coding variants in the *SCN5A* gene [16,17], which still remains the only gene with definitive evidence of an association with BrS and is clinically actionable [18,19,20]. Currently, more than 20 candidate BrS genes have been proposed [6,17,21,22,23,24,25,26,27,28,29,30,31,32,33,34,35,36,37,38,39,40,41,42,43], but their causality in BrS pathogenesis is widely debated [15]. The current knowledge about genetics in BrS is summarized in Table 1. To date, however, most patients do not carry causative mutations on the panel of BrS genes, highlighting the need for a better characterization of the molecular basis of this disorder.

The limited number of BrS cases with a clear monogenic inheritance has pointed toward new hypotheses of a more complex genetic architecture of the disease, involving multigenic inheritance and a polygenic risk score that can influence penetrance and risk stratification [44]. Recently, studies exploiting the genome-wide association study (GWAS) approach suggested that common genetic variations can modulate the phenotypic expression of BrS, providing evidence for a model of inheritance more complex than previously thought [17,43].

Indeed, polygenic risk score analyses based on several susceptibility variants demonstrate a cumulative contribution of common risk alleles among different BrS patients, as well as genetic associations with cardiac electrical traits in the general population, thus supporting the concept of “genomic arrhythmia” [43].

Moreover, the recent findings also highlight that genes encoding structural proteins or cardiac transcription factors are associated with the BrS phenotype, thus strengthening the hypothesis of overlap with structural cardiomyopathies [43,45].

Clinical BrS manifestations are more common in adults, and despite autosomal inheritance, they are eightfold more frequent in males than in females [4]. To date, gender differences in BrS phenotype manifestation are widely recognized: female patients less frequently display a type 1 Brugada ECG pattern and exhibit lower inducibility rates. But the underlying causality remains unclear and needs to be further investigated [46]. Recently, a higher prevalence of pathogenic variants in *SCN5A* has been published in symptomatic female patients with BrS compared with male patients, and an even higher prevalence in females with BrS with arrhythmic events [47] suggesting that pathogenic variants in *SCN5A* in women may be a risk factor, perhaps by overcoming a “protective” environment [1].

Overall, although different genetic approaches have been adopted, the characterization of BrS molecular bases remains limited. The identification of new candidate genes and risk factors can lead to a better definition of BrS pathogenic mechanisms, allowing an increase in diagnostic sensitivity and the improvement of family and clinical management and risk stratification.

## 3. Imaging Abnormalities

Cardiomyopathies are uniformly characterized by the identification of either structural or functional myocardial abnormalities via imaging techniques. Although most patients with BrS display no remarkable alterations on a transthoracic echocardiogram (TTE) or via cardiac magnetic resonance (CMR) imaging [1], some ECG findings have been suggested as possible indicators for underlying anatomical arrhythmogenic substrates [48] (Table 2). For instance, a correlation between patients with a spontaneous type 1 ECG pattern and a lower right ventricular ejection fraction (RVEF) has been described [49], as well as focal mechanical abnormalities in the RVOT [50].

Although the classical echocardiography parameters have a limited yield in BrS, new techniques including strain and speckle tracking [65,66] have led to a more accurate evaluation of the systolic and diastolic functions in BrS. In addition, the TEI index, which evaluates both systolic and diastolic time intervals to assess global cardiac dysfunction, has been used to differentiate BrS and non-BrS patients through a sodium channel blocking test: only the former ones showed prolonged PQ intervals and a decreased biventricular function at the TEI index [51,52]. Evaluation of the RV longitudinal strain with 2D speckle tracking quantifies regional myocardial deformation, with high spatial resolution speckle tracking not being affected by angle dependency or translation or tethering from the surrounding tissue [67]. The RV longitudinal strain has shown a significant reduction in BrS patients [51]. Moreover, speckle tracking echocardiography may help in differentiating BrS from the right bundle branch block (RBBB), as it was shown to track slower conduction through free wall segments which are found in RBBB but not in BrS [66].

CMR is an accurate and reproducible tool for estimating both left ventricular (LV) and RV volumes and is now considered the gold standard technique for cardiomyopathies [3,4]. Although controversial [68], anatomical involvement in BrS has been demonstrated in the literature. In detail, greater RV volumes and reduced RV function have been described [69], especially at RVOT [55]. In addition, some BrS patients display a midwall stria of late gadolinium enhancement within the LV consistent with an underlying cardiomyopathic process [69]. These findings lend further support to the presence of subtle structural abnormalities in BrS, with a possible evolution toward a cardiomyopathic phenotype over time [69]. Examples are shown in Figure 2. Additional morphofunctional abnormalities were recently reported: for instance, a direct correlation was shown between the LV/RV dilation and *SCN5A* mutation, with wider involvement of the RV than the LV [54] as observed in the classic arrhythmogenic right ventricular cardiomyopathy (ARVC). A recent study [70] allowed a more accurate localization of the aforementioned abnormalities, which appear to be more extended than RVOT as the ajmaline test had them localized both in the upper anterior wall but also in the antero-inferior wall, leading to an increased arrhythmogenic risk. In the same study, a significant correlation was observed between the RV dilation/dysfunction and *SCN5A* mutations [70]. In particular, the regional RV contractility abnormalities were found to be dynamic and functionally related to the expansion of the electrical substrate after ajmaline [70], accounting for the limited diagnostic value of baseline CMR.

## 4. Histopathological Findings

From the first characterizations of the ECG pattern, structural alterations, such as fibrosis, fibrolipomatosis, and RV cardiomyopathic changes, were described in patients with apparent idiopathic ventricular fibrillation (VF) [53]. As recently described for desmoplakin cardiomyopathy [71], cardiac inflammation might represent a “hot-phase” in BrS and lead to the natural progression of the disease [72,73].

Lymphocytic myocarditis (Figure 3) with inflammatory infiltrates and focal necrosis, with or without microaneurysms, was found in endomyocardial biopsies from the RV, and the LV as well, in patients with symptomatic BrS [56]. Among BrS patients, those who were carriers of *SCN5A* mutations displayed more cardiomyopathic changes. Remarkably, many patients were positive for intracardiac viral genomes. The authors suggest that the classic BrS ECG pattern is not a marker of a specific syndrome, but rather an electrical expression of RV structural abnormalities which may be the outcome of genetic, infective, and inflammatory conditions. In another study, RVOT endomyocardial biopsy, guided by a three-dimensional voltage map, showed that myocardial inflammation at histology correlated with a higher prevalence of abnormal bipolar map and greater bipolar low-voltage area extension in patients with BrS [10]. Notably, parvovirus B19 was associated with myocarditis-induced VF in many patients with BrS [57,58,59]. On the other hand, critical *SCN5A* variants can be found in patients with arrhythmic myocarditis, even in the absence of the BrS ECG pattern [74]. These findings support the role of myocardial inflammation as a possible arrhythmogenic substrate [75,76].

However, other studies failed to confirm definite myocarditis in biopsies from the RV, by showing only moderate myocardial hypertrophy, moderate fibrosis, and fatty replacement of the myocardium, with hypokinetic RV and RV trabeculae [77]. A genetically positive BrS patient who underwent a heart transplantation for recurrent VF episodes showed RV hypertrophy and fibrosis with epicardial fatty infiltration, which were deemed as the origins of ECG alterations. Specifically, the RVOT endocardium showed activation slowing due to interstitial fibrosis and was the origin of VF, without a transmural repolarization gradient, and with normal conduction in the LV [60]. Another patient with compound heterozygosity for a nonsense and a missense mutation in SCN5A revealed changes consistent with a dilated cardiomyopathy and advanced degeneration of the electrical conduction system with severe sodium channel dysfunction [78]. Even asymptomatic family members with BrS and *SCN5A* gene mutation showed histological abnormalities [79], and up to 33% of the families of patients suffering from unexplained sudden cardiac deaths with idiopathic fibrosis and/or hypertrophy received a post-mortem diagnosis of BrS [80].

Epicardial surface and interstitial fibrosis were described in BrS, along with increased collagen throughout the heart and a reduction in the expression of gap junctions in the RVOT. There was a correlation between structural abnormalities and abnormal potentials, and their ablation abolished the BrS phenotype and malignant arrhythmias [9]. Another group confirmed that BrS is associated with increased collagen content throughout the RV and the LV, but irrespective of sampling location or myocardial layer in patients experiencing sudden cardiac death [61]. Based on the data provided above, an endomyocardial biopsy could become a new diagnostic tool for the research of concealed morphological abnormalities in BrS, as well as for the identification of dynamic arrhythmogenic substrates [75,81,82].

## 5. Electroanatomical Substrates

Electroanatomical mapping (EAM) is an invasive method to visualize intracardiac electrical activation [83]. Low voltage and prolonged or fragmented ventricular signals reflect the arrhythmogenic substrate in BrS patients undergoing EAM [84] (Figure 4). Initial studies with endocardial mapping localized the electroanatomical substrate in the RVOT [10,84,85]. However, recent studies could demonstrate that the electroanatomical substrate is located most often on the epicardial surface of the RVOT [62,63,64].

In their landmark paper, Nademanee et al. were the first group that performed endocardial and epicardial mapping of the RVOT in a case series of nine patients with a type 1 BrS ECG pattern. They demonstrated that the underlying mechanism is delayed depolarization over the anterior aspect of the RVOT epicardium [63]. The issue is relevant since the RVOT has distinct electrophysiological properties as compared to the surrounding myocardium [86]. Catheter ablation of the substrate resulted in the normalization of the BrS ECG pattern and the non-inducibility of VF/ventricular tachycardia (VT) in most patients [63]. Furthermore, ablation was associated with an event-free follow-up of 20 ± 6 months in all patients [63]. The largest study of endocardial and epicardial EAM with subsequent ablation in BrS patients (n = 135) was performed by Pappone et al. [64]. Combined endo-epicardial mapping localized the substrate exclusively on the anterior RVOT and RV anterior free wall of the epicardium. Ajmaline administration increased the area of the epicardial substrate and catheter ablation resulted in the normalization of the type 1 BrS ECG pattern and non-inducibility of VT/VF [64]. The substrate size correlates with the arrhythmia inducibility during the electrophysiologic study [87]. A cutoff of > 4 cm^2^ of the abnormal electrophysiological substrate on EAM was described as an independent predictor of inducible ventricular arrhythmias (VT/VF) during programmed ventricular stimulation [88]. Radiofrequency catheter ablation of ventricular arrhythmias can reduce the burden of VT/VF and is now recommended for patients with recurrent ICD shocks or patients who are not suitable or decline an ICD according to current US guidelines (class I indication, level of evidence B from non-randomized trials) [4]. In the recent HRS/EHRA/APHRS/LAHRS expert consensus statement [83], catheter ablation was assigned a class IIa indication (level of evidence B from non-randomized trials) for patients with recurrent sustained ventricular arrhythmias or implantable cardioverter defibrillator (ICD) therapies. The ablation strategy has shifted away from targeting premature ventricular complex-triggered VF in BrS patients [89] and toward directly targeting the substrate on the epicardial aspect of the RVOT [90]. In a systematic review of 233 patients from 11 case series and 11 case reports, it has been demonstrated that endocardial mapping alone does not identify the electroanatomic substrate in 93% of cases and that epicardial substrate modification via catheter ablation is more effective than an endocardial-only approach [90].

## 6. Critical Review of Arrhythmic Risk Stratification

The identification of high-risk BrS patients remains a pivotal issue for the prevention of sudden cardiac death (SCD). Although almost every author agrees on the importance of symptoms and a spontaneous type 1 pattern [3,4], some other risk markers are controversial. An aborted SCD or documented VT/VF are clear recommendations [3] for ICD implantation, giving the burden of recurrences as high as 8–10% per year [91]. Patients diagnosed after syncope are still at a high risk irrespective of a spontaneous ECG pattern (1.9% per year if a type-1-induced pattern vs. 2.3% per year if a spontaneous type 1, statistically nonsignificant), provided vasovagal etiology has been excluded [92]. Defining the etiology of every syncope is often challenging, and a great effort should be directed toward history collection to improve patient selection for ICD implants.

A spontaneous type 1 BrS pattern is consistently associated with a higher event rate, even when asymptomatic (1.2% per year vs. 0.4% per year in drug-induced type 1, *p* = 0.049) [92]. However, since the longitudinal variation in the ST-segment in the right precordial leads is well described [93,94], a structured follow-up must be considered by employing 12-lead ECG Holter recordings [95].

Considering the high psychological and physical impact of an ICD in a young population, in the last two decades, other features have been proposed for better stratification of the arrhythmic risk in BrS. So far, little evidence corroborates the hypothesis of a strong association between a specific gene mutation and a worse prognosis [1]. Therefore, multimodal prognostic workup should also include clinical, electrocardiographic, and electrophysiological parameters.

Among the clinical parameters, age and the number of familial cases of SCD can help to define the individual risk [96,97]. Although few data are available for younger (<12 years old) and older (>60 years old) BrS patients, the event risk seems lower in elderly patients [98]. Males are largely predominant in all BrS groups, including SCD and syncope patients, driving a threefold increase in event risk. For female patients, a PQ interval greater than 200 ms as well as sinus node dysfunction have been proposed as risk factors [99], defining a strong role for the hormonal balance on the sodium channels pathophysiology.

As for the electrocardiographic parameters, QRS fragmentation and duration [100], late potentials [101], and the aVR sign [97] have been proposed. In particular, QRS fragmentation is associated with a twofold to ninefold increase in events, depending on filtering and recording modalities in different studies [3]. Furthermore, the aVR sign [97] establishes a link with the pathophysiology of the disease, analyzing right ventricle outflow tract involvement in some severe arrhythmic phenotypes. The extent of the ST alterations was in turn linked to the severity of the arrhythmic risk. A BrS type 1 pattern in the peripheral leads [102] and early repolarization pattern in the inferior leads [103] were linked to an increased arrhythmic risk. Instead, a prolonged (>200 ms) T peak–T end interval was not confirmed in different studies [104,105]. A recent paper focused on the depolarization delay shown by the r’ wave morphology. The authors found a strong correlation between the dST-Tiso interval and the VT/VF inducibility during the EPS [106]. Nevertheless, further evidence is needed to use this marker as an independent risk factor.

Albeit controversial, the prognostic value of VF/VT induction during the electrophysiological test (EPS) remains a cornerstone in clinical practice. The latest ESC guidelines [3] assign a class IIb to the ICD implantation after a positive EPS in asymptomatic patients with a spontaneous type 1 BrS ECG. No universal agreement exists about the stimulation protocol, but a standardized right apex and outflow tract stimulation, a drive train [S1-S1] of eight beats at 600 and 400 ms, and three extrastimuli [S2-S3-S4] with a minimum coupling interval of 200 ms have been suggested [107]. Inducibility is confirmed if sustained VT or VF are recorded [107]. Nonetheless, the association between a positive EPS and subsequent clinical events was confuted by many authors [108,109]. Metanalysis and large observational studies [110,111] found a more than twofold (hazard ratio 2.55) augmented risk of spontaneous VF after a positive EPS. Furthermore, an induction with a single extrastimulus or a ventricular refractory period < 200 ms [3] are valuable elements of vulnerability to consider in a multiparameter assessment.

An overview of the prognostic factors for BrS is provided in Table 3. Given the multiple and controversial candidate prognostic factors, different risk scores have been proposed in the last few years to improve the SCD risk stratification in BrS [5,112]. However, these scores showed good results for low- and high-risk patients, but poor performances in the large grey zone of the intermediate-risk patients [113]. However, since most evidence to date is based on small case series or isolated reports, it is still not possible to define a hierarchy of prognostic factors as well. Dedicated studies with uniform design and advanced diagnostic workup are needed for improving it.

In this setting, the identification of novel prognostic signs from concealed structural abnormalities may considerably improve the patient selection for ICD implants, especially with a primary prevention indication.

## 7. Conclusive Remarks and Future Directions

We hereby showed that BrS sometimes displays concealed substrates that may be identified via advanced diagnostic techniques, including CMR, EMB, and EAM. In this setting, diagnostic criteria for cardiomyopathy may be met more frequently than expected following genetics and simple tests such as ECGs. Waiting for further research and big data analysis, many parameters derived from advanced myocardial imaging, electroanatomical mapping, and histology may be included in a multimodal score to significantly improve the arrhythmic risk stratification of BrS. For instance, LGE [114], replacement fibrosis [115], low-voltage areas [116], and myocardial inflammation [117] are recognized risk factors for many cardiomyopathies. In the setting of BrS, major beneficial effects are expected from a multimodal assessment, in particular for the majority of patients currently classified at intermediate risk for SCD [113] and with no clear indications for a primary prevention ICD implant [3,4].

## Figures and Tables

**Figure 1 genes-13-01755-f001:**
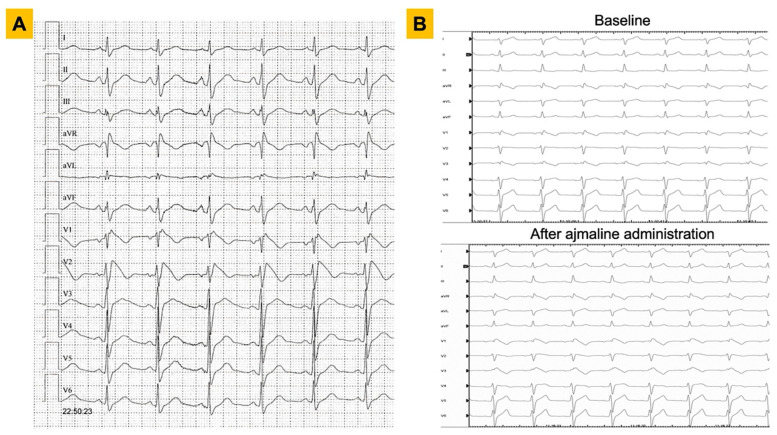
ECG findings in patients with Brugada syndrome. Representative examples of diagnostic ECGs for Brugada syndrome are shown. Panel (**A**). A 25-year-old male with a spontaneous type 1 pattern on 12-lead ECG. Panel (**B**). A 36-year-old male with right bundle branch block pattern on baseline ECG (upper panel) and subsequent unmasking of a type 1 Brugada ECG pattern after administration of ajmaline at 1 mg/kg (lower panel). ECG = electrocardiogram.

**Figure 2 genes-13-01755-f002:**
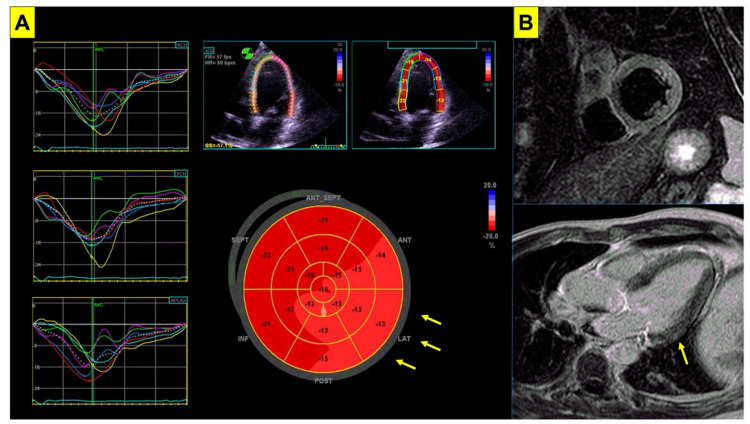
Imaging abnormalities in Brugada syndrome. Subtle imaging abnormalities associated with BrS are shown. Panel (**A**) echocardiogram of a patient with genetically proven BrS. Despite normal left ventricular systolic function (LVEF = 62%), impairment in global longitudinal strain is shown (GLS = −16%, nv < −20%) mainly involving the lateral wall (arrows). Panel (**B**) cardiac magnetic resonance in the same patient shows slight hyperintensity in T2-weighted short tau inversion recovery sequences (STIR, upper panel) involving the inferolateral basal segment of the left ventricular wall, and focal late gadolinium enhancement (LGE, lower panel) involving the basal segment of the lateral wall (arrow). BrS = Brugada syndrome; LVEF = left ventricular ejection fraction.

**Figure 3 genes-13-01755-f003:**
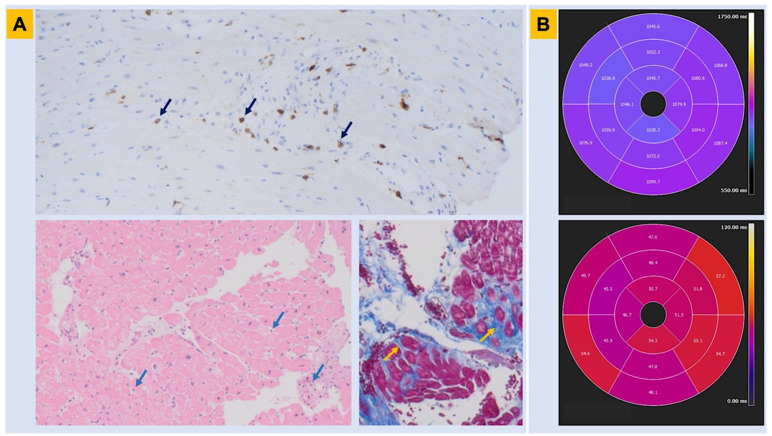
Histopathologic findings in Brugada syndrome. Myocardial tissue abnormalities associated with BrS are shown. Panel (**A**). Endomyocardial biopsy obtained from the right ventricle in a patient with genetically proven BrS shows lymphocytic inflammatory infiltrate with a CD3+ T-cell count consistent with myocarditis (immunohistochemistry assay—upper panel; hematoxylin-eosin assay—lower left panel; arrows). In the same patient, trichrome assay identifies areas of interstitial and replacement-type fibrosis, in blue color (lower right panel, arrows). Panel (**B**). Cardiac magnetic resonance obtained in the same patient before the automated cardioverter defibrillator implant shows abnormalities in parametric mapping involving the inferolateral left ventricular wall in both T1 and T2 sequences (n.v. for parametric mapping: T1 < 1045 ms; T2 < 50 ms). BrS = Brugada syndrome.

**Figure 4 genes-13-01755-f004:**
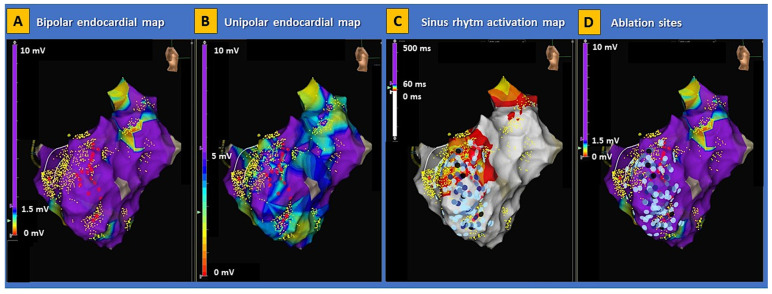
Electroanatomical substrate of Brugada syndrome. Examples of electroanatomical map abnormalities involving the right ventricle are shown in a patient with genetically proven BrS. The disproportion between low-voltage areas in bipolar (panel (**A**)) and unipolar (panel (**B**)) endocardial maps indicates the presence of a deep arrhythmogenic substrate consistent with classic arrhythmogenic right ventricular cardiomyopathy. The activation map during sinus rhythm (panel (**C**)) shows an extensive area of late potentials within the basal lateral segment of the right ventricle. In this patient, radiofrequency energy was extensively delivered (panel (**D**)), aimed at the complete abolishment of abnormal potentials in the right ventricle. No ventricular arrhythmias were induced on post-procedural programmed ventricular stimulation. BrS = Brugada syndrome.

**Table 1 genes-13-01755-t001:** Genetics of Brugada syndrome.

BrS Susceptibility Genes		Prevalence in BrS Cases	BrS Risk Loci	SNPs
*SCN5A*		20–25%	*SCN5A*	rs11708996
				rs7638909a
				rs62241190a
				rs7374540a
				rs7433206a
				rs34760424a
				rs41310232a
				rs6782237a
*SCN10A*		>5%	*SCN10A*	rs10428132
				rs6801957
*CACNA1C*	*CACNB2*	1–2%	*HEY2*	rs9388451
				rs9398791
*PKP2*	*HCN4*	<1%	*HDDC2*	rs6913204a
*KCNH2*	*KCNE3*			rs6913204a
*CACNA2D1*	*KCNJ8*		*TBX20*	s11765936
*KCND3*	*RANGFR*			rs340398a
*SCN2B*	*SCN1B*		*GATA4*	rs804281
*KCND2*	*TRPM4*		*ZFPM2*	rs72671655
*KCNE5*	*ABCC9*		*WT1*	rs72905083
*SCN3B*	*SLMAP*		*TBX5*	rs883079
*FGF12*	*SEMA3A*		*IRX3*	rs11645463
*GPD1L*			*IRX5*	rs72622262
References: [6,17,21,22,23,24,25,26,27,28,29,30,31,32,33,34,35,36,37,38,39,40,41,42,43]			*PRKCA*	rs12945884
			*MAPRE2*	rs476348
			*MYO18B*	rs133902

The current knowledge about genetics in BrS is shown, including susceptibility genes, prevalence, risk loci, and polymorphisms. BrS = Brugada syndrome; SNPs = single nucleotide polymorphisms [6,17,21,22,23,24,25,26,27,27,28,29,30,31,32,33,34,35,36,37,38,39,40,41,42,43].

**Table 2 genes-13-01755-t002:** Concealed substrates in Brugada syndrome.

Structural Findings for Underlying Anatomical Arrhythmogenic Substrates in BrS.	Studies
Echocardiogram - Decreased biventricular function at TEI index through sodium channel blocking test. - Reduction in the RV longitudinal strain. - Morphologic and wall-motion abnormalities of the RV.	[51] [52] [53]
Cardiac magnetic resonance - Lower right ventricular ejection fraction. - LV/RV dilation, with a wider involvement of the RV than LV. - Enlarged RVOT area, larger RV end-systolic volumes, lower LV and RV ejection fraction. - Fibrosis and abnormal late fractionated potentials, indicative of slowed conduction in the RVOT region.	[49] [54] [55] [9]
Endomyocardial biopsy - Lymphocytic myocarditis with inflammatory infiltrates and focal necrosis, with or without microaneurysms. - Parvovirus B19 with VF. - RV hypertrophy and fibrosis, mainly at RVOT level, with epicardial fatty infiltration. - Epicardial interstitial fibrosis, along with increased collagen throughout the heart and a reduction of the expression of gap junctions in the RVOT.	[56] [57] [58] [59] [60] [9] [61]
EAM - Electroanatomical substrate on the epicardial rather than endocardial surface of the RVOT.	[62] [63] [64]

Subtle substrate abnormalities with a potential arrhythmogenic role in BrS are shown, as documented by the multimodal diagnostic workup. BrS = Brugada syndrome; EAM = electroanatomical map; RV = right ventricular; RVOT = right ventricular outflow tract.

**Table 3 genes-13-01755-t003:** Known and candidate prognostic factors for Brugada syndrome.

Prognostic Factors Accounting for Arrthymogenic Risk in BrS.	Studies
ECG - Spontaneous type I Brugada pattern, even when asymptomatic. - PQ interval greater than 200 ms, as well as sinus node disfunction. - QRS fragmentation and duration. - Late potentials. - aVR sign. - Brugada type 1 pattern in the peripheral leads. - Early repolarization pattern in the inferior leads.	[92] [99] [100] [101] [97] [102] [103]
Genetics - SCN5A mutation	[14] [15]
Echocardiogram - Focal abnormalities localized in the upper anterior wall but also in the antero-inferior wall.	[70]
Endomyocardial biopsy - Myocardial inflammation. - Carriers of *SCN5A* mutations.	[10] [56]
Cardiac magnetic resonance - Late gadolinium enhancement.	[69]
EPS - Length of the dST-Tiso interval. - Induction of ventricular arrhythmias with a single extrastimulus or a ventricular refractory period <200 ms.	[106] [110] [111]
EAM - Area > 4 cm^2^ of abnormal electrophysiological substrate.	[88]

Factors, either known or potentially associated with an increased arrhythmic risk in BrS, are shown. BrS = Brugada syndrome; EAM = electroanatomical map; EPS = electrophysiological study.

## Data Availability

Not applicable.

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
