# Peer review of "Concealed Substrates in Brugada Syndrome: Isolated Channelopathy or Associated Cardiomyopathy?"

_genes, 2022, doi:10.3390/genes13101755_

Round 1
Reviewer 1 Report
In this review, the authors aim to support the cardiomyopathic nature of the BrS. I still have some concerns:
1. What’s the significance of defining BrS as a “channelopathy” or “cardiomyopathy”?
2. Currently, do BrS patients also need to do additional tests to determine the potential cardiomyopathy?
3. Do all the BrS patients with arrhythmia have structural or functional myocardial abnormalities?
4. What are the direct factors that contribute to the arrhythmia in BrS? Is myocardial inflammation the most important factor?
5. Which is primary and which is secondary in BrS, the arrhythmias or the structural/functional myocardial abnormalities?
Reviewer 2 Report
I understand an imperative to see possible causal factors in Brigada Syndrome (BS) based on an observation that some patients with BS have additional CV issues such as fibrosis. However, to float the hypothesis that the channelopathy is not the cause of BS requires some very hard evidence:
1. First, if the idea is that NaV channelopathy is sufficient but not necessary for BS, you need to document cases of genuine BS where there is no channelopathy. Can you do this?
2. If not then you must abandon the notion of Nav channelopathy sufficient but not necessary.
3. This means that the channeopathy is necessary. In which case the comorbidities could either be coincidence or the consequence of the channelopathy. This may be a topic worthy of discussion
4. If so (see Q1) you need to consider the definition of BS. Because if BS can occur without channelopathy you next need to consider whether BS can occur when there is only channelopthay. This impacts on the definition of BS and may also be worthy of discussion.
If you fail to approach the topic clearly, with an ‘if, then, else’ approach, the article will degenerate into little more than speculation. This would be potentially dangerous. Let’s see how you have approached this….
Early on you note that most BS patients are ‘without a positive genetic diagnosis’. What does this mean? Does it mean that no channelopathy has been found or that it has been found there is no channelopathy? How is this compatible with your opening statement that “Brugada syndrome [BrS] is an inherited autosomal dominant genetic disorder”. Either it is or it isn’t!
You then define BS as a syndrome with an arrhythmia: “Current international guidelines [3,4] agree in defining BrS in presence of a type-1 Brugada electrocardiogram [ECG] pattern, i.e. a persistent ST-segment elevation ≥ 2 mm followed by a negative T-wave in ≥ 1 of the right precordial leads V1 to V2, occurring either spontaneously or following a sodium channel blocker test [Figure 1]. However, according to the Shanghai score system of 2016 [5], if the type-1 pattern is unmasked during a sodium channel blocker test, then clinical history, family history and genetic test need to be evaluated to meet the diagnostic criteria.”. Well this isn’t science. Tis is about clinical medicine and being sure that a diagnostic mistake has not been made.
So what is BS? A syndrome with an arrhythmia of a certain type, or a syndrome with a sodium channelopathy? You are already in danger of invoking unclear criteria to define a syndrome and then (presumably) arguing that the diagnostic criteria are not necessarily causative in the clinical manifesto. It is not scientifically appropriate to pursue such a narrative.
You go on to say “Despite recent advances, the pathogenetic mechanisms of the disease remain to be fully understood.”. This is an entirely different issue. However with the foundation for the narrative so weak, the article is inevitably going to descend into a search for statistical correlations.
I’m sorry but you have not made a strong enough case for this article. It is not helpful and might even be a source of unfounded distraction.
The rest of the article is a list of comorbidities in BS patients, without any attempt to disentangle cause and effect, as I feared.
Round 2
Reviewer 1 Report
The authors have addressed my concerns.
Reviewer 2 Report
Thank you for revising your article. You have clarified your points and taken the article more deeply into the realm of clinical medicine. At this point I must withdraw from further engagement because I am not a clinician. As far as I am concerned the paper is acceptable.